# Effect of the Gintonin-Enriched Fraction on Glucagon-Like-Protein-1 Release

**DOI:** 10.3390/molecules26206298

**Published:** 2021-10-18

**Authors:** Rami Lee, Sun-Hye Choi, Han-Sung Cho, Hongik Hwang, Hyewhon Rhim, Hyoung-Chun Kim, Sung-Hee Hwang, Seung-Yeol Nah

**Affiliations:** 1Ginsentology Research Laboratory, Department of Physiology, College of Veterinary Medicine, Konkuk University, Seoul 05029, Korea; rmlee12@konkuk.ac.kr (R.L.); vettman@naver.com (S.-H.C.); newlove0202@nate.com (H.-S.C.); 2Center for Neuroscience, Korea Institute of Science and Technology, Seoul 02792, Korea; hongik.kist@gmail.com (H.H.); e-hrhim@kist.re.kr (H.R.); 3Neuropsychopharmacology and Toxicology Program, College of Pharmacy, Kangwon National University, Chunchon 24341, Korea; kimhc@kangwon.ac.kr; 4Department of Pharmaceutical Engineering, College of Health Sciences, Sangji University, Wonju 26339, Korea; d639sunghee@hotmail.com

**Keywords:** ginseng, gintonin, gintonin-enriched fraction, NCI-H716 cell, GLP-1 secretion

## Abstract

Ginseng-derived gintonin reportedly contains functional lysophosphatidic acids (LPAs) as LPA receptor ligands. The effect of the gintonin-enriched fraction (GEF) on in vitro and in vivo glucagon-like protein-1 (GLP-1) secretion, which is known to stimulate insulin secretion, via LPA receptor(s) remains unclear. Accordingly, we examined the effects of GEF on GLP-1 secretion using human enteroendocrine NCI-H716 cells. The expression of several of LPA receptor subtypes in NCI-H716 cells using qPCR and Western blotting was examined. LPA receptor subtype expression was in the following order: LPA6 > LPA2 > LPA4 > LPA5 > LPA1 (qPCR), and LPA6 > LPA4 > LPA2 > LPA1 > LPA3 > LPA5 (Western blotting). GEF-stimulated GLP-1 secretion occurred in a dose- and time-dependent manner, which was suppressed by cAMP-Rp, a cAMP antagonist, but not by U73122, a phospholipase C inhibitor. Furthermore, silencing the human LPA6 receptor attenuated GEF-mediated GLP-1 secretion. In mice, low-dose GEF (50 mg/kg, peroral) increased serum GLP-1 levels; this effect was not blocked by Ki16425 co-treatment. Our findings indicate that GEF-induced GLP-1 secretion could be achieved via LPA6 receptor activation through the cAMP pathway. Hence, GEF-induced GLP secretion via LPA6 receptor regulation might be responsible for its beneficial effects on human endocrine physiology.

## 1. Introduction

Ginseng (*Panax ginseng* C.A. Meyer, family Araliaceae) has long been used as a tonic herbal medicine to enhance the quality and duration of life. It contains a diverse range of bioactive compounds, including well-known ginsenosides, acidic polysaccharides, and the recently identified gintonin [1]. Ginseng possesses a wide range of therapeutic properties, including anti-aging, anti-fatigue, antioxidant, vasodilatory, anti-inflammatory, antidiabetic, and antipsychotic effects [1,2,3,4,5].

Gintonin is a newly identified non-saponin/non-acidic ingredient isolated from ginseng. Lysophosphatidic acids (LPAs) are one of the functional components of gintonin. There are six well-known LPA receptor subtypes [6], and gintonin LPAs are considered exogenous ligands of these six G-protein-coupled LPA receptor subtypes [7]. Gintonin induces LPA receptor activation in neuronal or non-neuronal cells through pertussis toxin-sensitive and -insensitive G proteins to release intracellular calcium. Given the mode of action, gintonin stimulates the release of Ca^2+^-dependent gliotransmitters, neurotransmitters, and hormones [8,9,10,11,12].

Accumulating evidence has revealed the in vivo effects of long-term oral administration of gintonin on the pathophysiology of conditions such as Alzheimer’s disease, Parkinson’s disease, and Huntington’s disease and that these effects are mediated via the LPA1 receptor [8,10,13,14,15,16,17]. To date, physiological/pharmacological studies assessing gintonin primarily focused on degenerative brain diseases, neglecting potential actions on other organs. We recently reported that gintonin suppresses depressive-like behavior by stimulating gut enterochromaffin cells to secrete serotonin [18]. However, the in vitro and in vivo effects of gintonin on hormonal release from other endocrinological cells in the intestines remain unknown.

The L cell is a representative intestinal enteroendocrine cell localized in the human intestinal colon [19]. It is well established that nutrient intake and ingestion in the human intestine are vital for maintaining metabolic homeostasis. Notably, the lipid fraction of gintonin is absorbed in the human intestine [10]. L cells are known to release glucagon-like peptide-1 (GLP-1, molecular weight (MW) = 3.3 kDa) [20]. As a physiological regulator, GLP-1 activates insulin secretion via glucose stimulation and inhibits pancreatic β-cell apoptosis, along with other diverse functions [21]. GLP-1 secretion starts with food arrival in the stomach or following the activation of the parasympathetic nervous system [22]. L cells, which are widely expressed in the colon, are mostly associated with energy metabolism. Recent studies have revealed that the human NCI-H716 cell line, derived from human colon L cells, can be employed as a model cell for GLP-1 secretion [20].

Furthermore, a few LPA receptor subtypes (LPA1, -2, -5, and -6) are expressed in the mouse intestine [23,24,25]. However, the relationship between LPA receptor subtypes and physiological GLP-1 secretion in human NCI-H716 cells remains undetermined. Accordingly, we utilized the human NCI-H716 cell line to elucidate the mechanism underlying gintonin-mediated GLP-1 release, both in vitro and in vivo.

In the present study, the expression of LPA receptor subtypes was firstly evaluated to detect that the LPA6 receptor was predominantly expressed in human NCI-H716 cells. In addition, we observed that the gintonin-enriched fraction (GEF) stimulated GLP-1 secretion via the LPA6 receptor through the cAMP pathway. Furthermore, orally administered GEF increased the levels of serum GLP-1 in mice. Finally, we discuss the physiological functions associated with in vitro and in vivo GEF-mediated GLP-1 secretion.

## 2. Results

### 2.1. Profiles of LPA Receptor Subtype Expression in Human NCI-H716 Cells

To date, six LPA receptor subtypes have been identified and described [1,26]. Accordingly, we first determined the expression levels of LPA receptors in human NCI-H716 cells. Real-time polymerase chain reaction (PCR) was performed to quantify the transcript-level expression of target genes in NCI-H716 cells. The expression of LPA6 was approximately 100-fold higher than that of LPA1, which was set at one. The average relative mRNA expression levels were in the following order: LPA6 > LPA2 > LPA4 > LPA5 > LPA1 (Figure 1a). Then, Western blotting was performed to determine the relative protein expression of LPA receptor subtypes. LPA6 exhibited the highest expression when compared with that of LPA1 (set at one) (Figure 1b). LPA3 and LPA5 showed lower expression than the basal level, which was set at one. The relative protein expression levels were in the following order: LPA6 > LPA4 > LPA2 > LPA1 > LPA3 > LPA5 (Figure 1b). This indicated the predominant expression of the LPA6 receptor subtype in human NCI-H716 cells.

### 2.2. Stimulatory Effects of GEF on GLP-1 Secretion

We examined whether GEF stimulated GLP-1 secretion in NCI-H716 cells. NCI-H716 cells treated with GEF at 0, 0.1, 0.3, 1, 3, 10, 30, and 100 µg/mL showed dose-dependent GLP-1 secretion for 1 h. High concentrations of GEF (10, 30, and 100 µg/mL) significantly stimulated GLP-1 secretion (Figure 2a). As reported by Kim et al. (2014), carbohydrate digestion increases glucose levels; hence, we utilized glucose as a positive control [27]. In addition, LPA (5 µM) was used as a positive control to stimulate GLP-1 secretion as it is the main component of GEF (Figure 2a). Notably, GEF did not affect the survival of NCI-H716 cells (Appendix A).

We next examined whether GEF affected GLP-1 secretion in NCI-H716 cells over a predetermined time course. A time-dependent manner of GEF (30 µg/mL)-stimulated GLP-1 secretion was observed in NCI-H716 cells (Figure 2b). Cells treated with glucose were used as a positive control and showed a significant increase in GLP-1 release. Interestingly, short-term exposure to GEF (30 min) did not induce GLP-1 secretion, while GEF treatment for 1 and 2 h significantly increased GLP-1 secretion (Figure 2b).

Next, we evaluated whether GLP-1 secretion was suppressed by antagonists that block the LPA receptor signaling pathway. First, cells were treated with Ki16425, an LPA1/3 receptor antagonist, in the absence or presence of GEF (100 µg/mL). We observed that Ki16425 treatment did not block GLP-1 release. In addition, U73122, a phospholipase C (PLC) inhibitor, did not suppress GEF-mediated GLP-1 secretion (Figure 2c), indicating that LPA1/3 receptor subtypes and the PLC pathway are not involved in mediating GLP-1 secretion (Figure 2c). Interestingly, GEF did not elicit [Ca^2+^]_i_ transients in NCI-H716 cells even at high concentrations (Appendix A), highlighting the possibility that GEF-mediated GLP-1 secretion is not related to the [Ca^2+^]_i_ transient.

Finally, as previous reports showed that GLP-1 secretion is stimulated by cAMP [28], we examined the involvement of the cAMP pathway. Accordingly, cell treatment with cAMP-Rp, a cAMP analog and a competitive antagonist of cAMP, was performed. Compared with the non-cAMP-Rp treatment group, cAMP-Rp treatment significantly blocked the GEF-mediated GLP-1 secretion in NCI-H716 cells, indicating that GEF-mediated GLP-1 secretion might be associated with the cAMP pathway (Figure 2d).

### 2.3. Effect of siRNA-Mediated LPA6 Knockdown on GEF-Induced GLP-1 Secretion

The next step aimed to confirm the involvement of LPA6 in GEF-induced GLP-1 secretion in NCI-H716 cells. Accordingly, downregulation of the expression of LPA6 was performed using the strategy of siRNA-mediated knockdown. Quantitative RT-PCR and Western blotting were employed to assess the extent of LPA6 knockdown using reverse-transcription PCR (RT-PCR). Quantitative RT-PCR and immunoblotting revealed that LPA6 was knocked down 48 h after transfection (Figure 3a,b). Based on in-house established methods, we employed 20 nM siRNA to determine the involvement of LPA6 in GEF-induced GLP-1 secretion. It was observed that 20 nM siLPA6 significantly attenuated the mRNA and protein expression of the LPA6 receptor subtype. Importantly, GEF-induced (10 and 30 µg/mL) GLP-1 secretion was significantly attenuated in siLPA6-transfected cells (Figure 3), indicating the possibility that GEF-induced GLP-1 secretion could be mediated via the LPA6 receptor subtype in human NCI-H716 cells.

### 2.4. Effect of GEF Intake on In Vivo GLP-1 Secretion

Next, it was examined whether oral administration of GEF increased serum GLP-1 levels. Accordingly, 50 and 100 mg/kg GEF (peroral (*p.o*.)) were administered to mice. Interestingly, we observed that GEF (50 but not 100 mg/kg) significantly increased serum GLP-1 levels at both 8 and 24 h, indicating that orally administered GEF elevates serum GLP-1 levels (Figure 4). We also observed that Ki16425 treatment did not block GLP-1 release.

## 3. Discussion

In the present study, it was first reported that six LPA receptor subtypes are expressed in a human intestinal model of L cells, i.e., NCI-H716 cells, demonstrating different degrees of expression. Among these, the LPA6 receptor subtype exhibited the highest expression levels. GEF, as an exogenous LPA receptor ligand, can stimulate in vitro and in vivo GLP-1 secretion. In addition, it was observed that GEF-mediated GLP-1 secretion was attenuated by silencing the LPA6 receptor subtype and cAMP-Rp, a competitive cAMP antagonist. Thus, it can be concluded that GEF-mediated GLP-1 secretion was not dependent on the [Ca^2+^]_i_ transient. Previously, our group reported that the GEF-mediated release of hormones and neurotransmitters via LPA1/3 receptors in neuronal cells is mediated via [Ca^2+^]_i_ transients [7]. These results suggest that GEF-mediated GLP-1 secretion is mediated via a novel pathway, such as LPA6-cAMP, rather than [Ca^2+^]_i_ transients.

GLP-1 is released in response to food intake to increase the action of incretin hormones and induce nutrient-induced insulin secretion [29,30,31]. It is well known that GLP-1 released from intestinal L cells has physiological functions such as delaying gastric emptying, reducing food intake, and enhancing antidiabetic actions [31,32,33,34]. Accordingly, GLP-1 has attracted considerable attention for developing anti-obesity or type 2 diabetes drugs using GLP-1 receptor agonists [35,36,37,38]. Some GLP-1 agonists are currently available to treat obesity [39].

Relatively little is known regarding the involvement of LPA and LPA receptor subtypes in GLP-1 secretion, although intestines express subsets of LPA2/5 receptor subtypes [25,40]. Herein, it was also observed that the LPA6 receptor is expressed at a higher level than LPA2/5 receptors (Figure 1). Thus, the human intestine possibly expresses the LPA6 receptor subtype, whose function(s) might be coupled to GLP-1 secretion via stimulation by endogenous or exogenous LPA in NCI-H716 cells (Figure 2a). Corroborating this notion, we observed that transfection of NCI-H716 cells with siRNA targeting the LPA6 receptor attenuated LPA6 receptor expression as well as GEF-induced GLP-1 secretion (Figure 3). These results suggest that the LPA6 receptor subtype contributes to GEF-stimulated GLP-1 secretion.

To further determine the downstream pathway of GEF-mediated GLP-1 secretion in NCI-H716 cells, U73122, Ki16425, and cAMP-Rp were employed (Figure 2). Our findings revealed that GEF-mediated GLP-1 secretion was not suppressed either by U73122, a PLC inhibitor, or Ki16425, an LPA1/3 receptor antagonist (Figure 2). In contrast, cAMP-Rp had an inhibitory effect on GEF-induced GLP-1 secretion in human NCI-H716 cells, indicating a relationship between GEF-induced GLP-1 release and cAMP pathways rather than the PLC pathway (Figure 2). A previous study has reported that activation of the LPA6 receptor subtype elevates cAMP levels in RH7777 cells transfected with the LPA6 receptor subtype [41]. This is consistent with a previous report that LPA6 receptor signaling is mediated via the cAMP pathway [41]. In addition, it was demonstrated that GEF-mediated GLP-1 secretion could be achieved via the LPA6-cAMP pathway, as cAMP antagonists can attenuate GEF-mediated GLP-1 secretion in NCI-H716 cells. Thus, the present and previous studies indicate that physiological effects of GEF can be achieved via two discrete second messenger systems, i.e., Ca^2+^ and cAMP, for neurotransmitter release or GLP-1 secretion, respectively (Figure 5) [41].

In addition, ginsenosides, also known as ginseng saponins, reportedly trigger GLP-1 release from NCI-H716 cells [27,42]. Although some ginseng saponins are related to GLP-1 secretion, there is no concrete explanation for the types of receptor(s) and downstream effectors involved in GLP-1 secretion mediated by ginseng saponins. Thus, the molecular mechanisms of GLP-1 secretion by ginseng saponins need to be further elucidated. Herein, human NCI-H716 cells are considered a useful in vitro tool to examine GEF-induced GLP-1 secretion from the human intestine. Given that the LPA6 receptor subtype is abundantly expressed in human NCI-H716 cells and that oral administration of GEF increased serum GLP-1 (Figure 4), further studies are required to elucidate the physiological function of GEF-mediated GLP-1 secretion from the human intestine. Interestingly, we found that although the in vitro effects of GEF on GLP-1 secretion were dose-dependent, GEF’s effect on in vivo GLP-1 secretion was higher at 50 mg/kg than at 100 mg/kg GEF. Similarly, compound K, a ginsenoside metabolite, exhibited in vitro anti-inflammatory effects in a dose-dependent manner by reducing inflammatory cytokines. However, in an in vivo study, oral administration of compound K did not show a dose-dependent effect on arthritis score in mice [43]. Currently, we cannot explain exactly why the low dosage of GEF exhibits better effects. One possible explanation based on the present results is that a low dosage of GEF might be saturable to stimulate GLP-1 secretion, whereas a high dosage of GEF might be too high to further stimulate GLP-1 secretion. Further study will be required to elucidate the discrepancy between in vitro and in vivo effects in GEF-mediated GLP-1 secretion in the future.

In summary, the modulation of GLP-1 secretion by LPA6 receptor signaling in human NCI-H716 cells and mice was demonstrated in this study. In the human NCI-H716 cell system, GEF enhanced the secretion of the enteroendocrine hormone GLP-1 in a dose- and time-dependent manner (Figure 2). Furthermore, the effect of GEF on GLP-1 secretion was inhibited by cAMP-Rp, indicating a cAMP-related signaling pathway mediated via the LPA6 receptor (Figure 5). Thus, GEF may exhibit beneficial effects on the human endocrine system through gut GLP-1 release.

## 4. Materials and Methods

### 4.1. GLP-1 Secretion from NCI-H716 Cells

Briefly, NCI-H716 cells were seeded at 0.5 × 10^6^ cells/well in 24-well microplates coated with Matrigel and incubated for 48 h. The cells were then washed twice with phosphate-buffered saline (PBS) and incubated with PBS containing either GEF (0, 0.1, 0.3, 1, 3, 10, 30, and 100 µg/mL), LPA (5 µM), or glucose (200 mM) at 37 °C for 1 h. For the inhibition of GLP-1 release, we utilized Ki16425 (10 µM) and U73122 (5 µM) with or without GEF (100 µg/mL). The supernatants were collected and either immediately assayed or frozen to later perform analysis using a GLP-1 active ELISA kit (Millipore, Billerica, MA, USA) at an excitation/emission wavelength of 355/460 nm. The values of secreted GLP-1 were corrected by measuring the amount of protein using the BCA protein assay.

### 4.2. siRNA Transfection in NCI-H716 Cells

A validated small interfering RNA (siRNA) targeting LPA6 (NP_001155969.1) was used. The validated siRNAs were MBS8230287-c (LPA6, siRNA) and MBS8241404 (negative control siRNA). The negative control siRNA was used to distinguish sequence-specific silencing from non-specific effects. The cells were transfected with siRNA using Lipofectamine RNAiMAX Reagent according to the manufacturer’s instructions, with modifications (Invitrogen, Van Allen Way Carlsbad, CA, USA).

### 4.3. Immunoblotting

A modified radioimmunoprecipitation assay buffer (RIPA buffer) supplemented with a protease inhibitor cocktail was employed to extract total protein from NCI-H716 cells and to evaluate the expression of LPA receptor subtypes (LPA1 to LPA6). Total protein (20 µg) was subjected to 10% sodium dodecyl sulfate polyacrylamide gel electrophoresis (SDS-PAGE) and blotted onto a 0.45-micrometer polyvinylidene difluoride (PVDF) membrane. Rabbit host-polyclonal antibodies, including anti-LPA1 antibody (Abcam, Cambridge, UK, ab23698, 1:2000), anti-LPA2 antibody (Abcam, Cambridge, UK, ab38322, 1:1000), anti-LPA3 antibody (Abcam, Cambridge, UK, ab23692, 1:500), anti-LPA4 antibody (Abcam, Cambridge, UK, ab203290, 1:2000), anti-LPA5 antibody (biorbyt, orb157371, Cambridge, UK, 1:2000), and anti-LPA6 (OriGene, AP52517PU-N, Rockville, MD, USA, 1:2000), were used for immunoblotting. The blotted membrane was stripped, washed three times, and re-probed with a horseradish-conjugated mouse anti-β-actin antibody (1:30,000). The PVDF membrane was visualized using Clarity Western ECL Substrate Bio-Rad (Hercules, CA, USA) on an iBright CL1000 imager (Thermo Fischer Scientific, Waltham, MA, USA).

### 4.4. Real Time PCR Analysis

The transcript-level expression of human LPA receptor subtypes (LPA1–6) was evaluated as described previously. Briefly, total RNA was isolated using TRIzol reagent (Invitrogen) and reverse-transcribed into cDNA. The reaction was performed on a CFX96 real-time PCR system (Bio-Rad, Hercules, CA, USA) using iQ SyBR Green Supermix (Bio-Rad, Hercules, CA, USA). The sequences of forward primers and reverse primers for the target genes were as follows: LPA1—5′gtcttctgggccattttcaa3′, 5′tcatagtcctctggcgaaca3′, 91 bp; LPA2—5′gaggccaactcactggtca3′, ggcgcatctcagcatctc3′, 58 bp; LPA3—5′gaagctaatgaagacggtgatga3′, 5′agcaggaaccaccttttcac3′, 135 bp; LPA4—5′tctggatcctagtcctcagtgg3′, 5′ccagacacgtttggagaagc3′, 107 bp; LPA5—5′cgccatcttccagatgaac3′, 5′tagcggtccacgttgatg3′, 66 bp; LPA6—5′tctggcaattgtctacccatt3′, 5′tcaaagcaggcttctgagg3′, 165 bp; β-actin (internal control)—5′ttctacaatgagctgcgtgtg3′, 5′ggggtgttgaaggtctcaaa3′, 122 bp.

### 4.5. Animal Experiment

In the present study, 6-week-old male ICR mice were purchased from Koatech (Pyeongtaek, Korea) and used for all experiments. Mice were randomly separated and housed under humidity levels of 50 ± 5% and a 12 h/12-h light and dark cycle. Water and food were provided ad libitum. The Guidelines for the Care and Use of Laboratory Animals of the Institute for Laboratory Animal Research (ILAR, 2010) were followed throughout the experimental process. This protocol was approved by the Institutional Animal Care and Use Committee of Konkuk University (No. KU21051-1). Mice were divided into three groups (*n* = 6 per group): the control group, the group receiving 50 mg/kg GEF (GEF50 group), and the group receiving 100 mg/kg GEF (GEF 100 group). The control group was treated with saline only, and the GEF50 and GEF100 groups were administered 50 or 100 mg/kg GEF, respectively. GEF was freshly dissolved in saline for all experiments. After GEF or Ki16425 administration, mice were euthanized by cervical dislocation, and blood samples were collected from the heart over a predetermined time course (0, 1, 2, 8, and 24 h). Cells were pretreated with Ki16425 before GEF administration (50 mg/kg). GEF was administered at 1, 2, 8, and 24 h, and 24 h was selected for Ki16425 administration. After being orally administered GEF, the mice were euthanized by cervical dislocation in consecutive order over the predetermined time course. Then, whole blood was immediately collected from the heart, and serum, treated with dipeptidyl peptidase-4(DPP-4) inhibitors, was frozen for later analysis.

### 4.6. Statistical Analysis

All measured and calculated values are presented as the mean ± standard deviation (SD). Statistical significance was set at *p* < 0.05. One-way analysis of variance or an unpaired Student’s *t*-test was used to compare differences between groups. In addition, multiple comparisons were performed using Dunnett’s method against the control group.

## Figures and Tables

**Figure 1 molecules-26-06298-f001:**
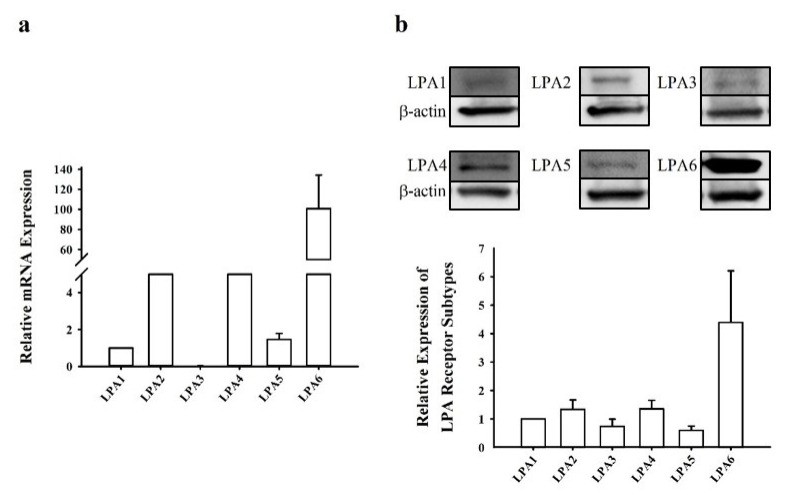
Expression levels of LPA receptor subtypes in NCI-H716 cells. (**a**) NCI-H716 cells were used to extract total RNA to screen the transcript-level expression of LPA receptor subtypes after 24 h from seeding the cells. LPA6 showed the highest expression, approximately 100-fold higher than that of LPA1, while LPA3 expression was not detected in this system. (**b**) Western blotting was used to screen the expression patterns of the LPA receptor subtypes after 24 h from seeding the cells. The expression of the LPA receptor subtypes was in the following order: LPA6 > LPA4 > LPA2 > LPA1 > LPA3 > LPA5. LPA, lysophosphatidic acid.

**Figure 2 molecules-26-06298-f002:**
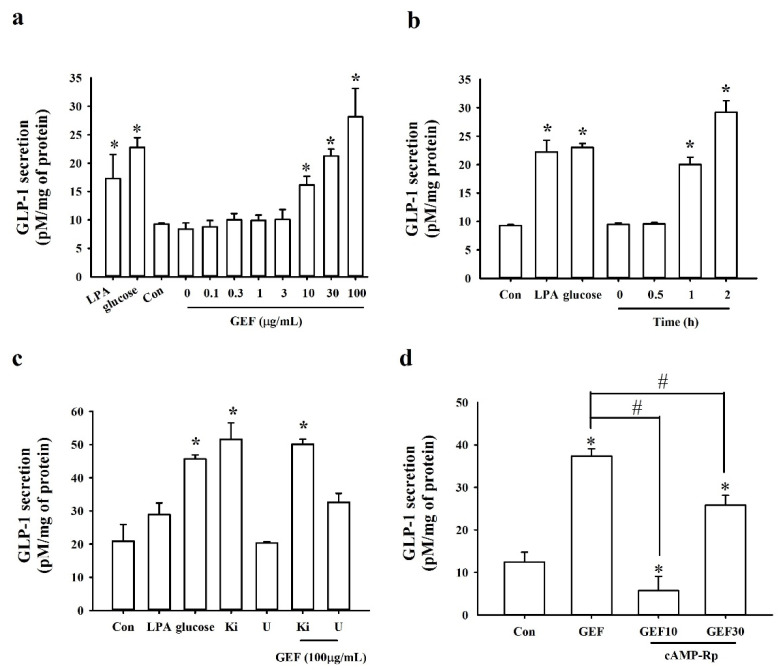
GEF-mediated GLP-1 secretion and inhibitory effect of antagonists in NCI-H716 cells. (**a**) GEF dose-dependently induces GLP-1 secretion. NCI-H716 cells were treated dose-dependently (GEF 0, 0.1, 0.3, 1, 3, 10, 30, or 100 μg/mL). (**b**) GEF-mediated GLP-1 release from NCI-H716 cells increases in a time-dependent manner. (**c**) GLP-1 secretion was not blocked by either U73122 (5 μM) or Ki16425 (10 μM). NCI-H716 cells were treated either with or without GEF (100 μg/mL) under antagonist treatment (U73122). (**d**) GEF-mediated GLP-1 secretion is blocked by cAMP-Rp (20 μM) under GEF treatment. Cell supernatant was collected and assayed with a GLP-1 active ELISA kit at an excitation/emission wavelength of 355/460 nm. * *p* < 0.05 vs. Control (Con), # *p* < 0.05 vs. Control (Con). GEF, gintonin-enriched fraction; GLP-1, glucagon-like protein-1; U, U73122 (5 μM); Ki, Ki16425 (10 μM); GEF10, GEF 10 μg/mL; GEF30, GEF 30 μg/mL.

**Figure 3 molecules-26-06298-f003:**
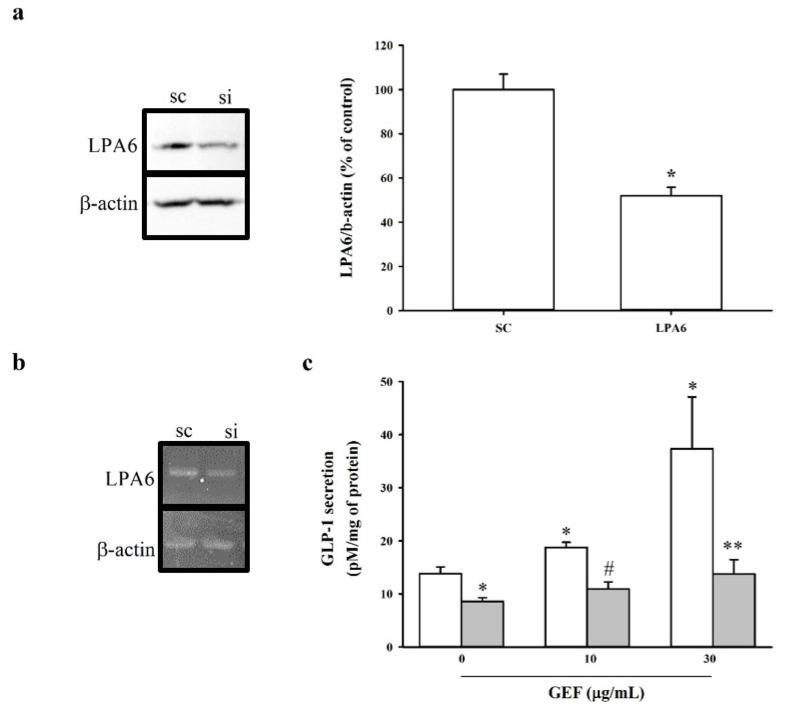
Effect of siLPA6 on LPA6 expression and GEF-induced GLP-1 secretion in NCI-H716 cells. NCI-H716 cells were transfected with either scrambled siRNA (sc) or siLPA6 (si) and incubated for 48 h. (**a**) Immunoblot analysis of LPA6 expression in NCI-H716 cells transfected with either scrambled siRNA (sc) or siLPA6. (**b**) qRT-PCR analysis of LPA6 expression in NCI-H716 cells transfected with either scrambled siRNA (sc) or siLPA6. Data represent means ± standard deviation (SD) (*n* = 3); * *p* < 0.5 sc vs. si. β-actin was used as a control. (**c**) Transfected NCI-H716 cells were treated with GEF (10 or 30 μg/mL) for 1 h. GLP-1 secretion was measured using a GLP-1 ELISA kit, as described in the Materials and Methods. Data represent means ± SD (*n* = 3); * *p* < 0.05 vs. sc (GEF 0 μg/mL), # *p* < 0.05 vs. sc (GEF 30 μg/mL), ** *p* < 005 vs. si (GEF 0 μg/mL). LPA6, lysophosphatidic acid receptor 6; GEF, gintonin-enriched fraction; GLP-1, glucagon-like protein-1.

**Figure 4 molecules-26-06298-f004:**
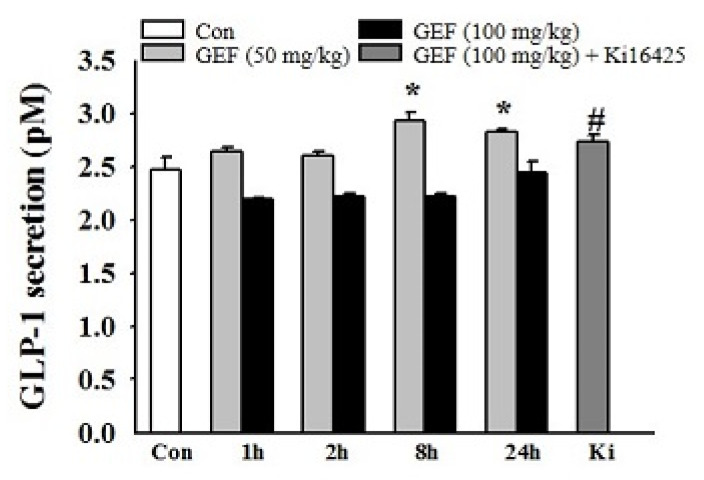
GEF-induced GLP-1 secretion in mice. GEF (50 or 100 mg/kg) was orally administered for the indicated periods (0, 1, 2, 8, or 24 h), followed by co-treatment with Ki16425 for 24 h. GLP-1 secretion was measured using a GLP-1 ELISA kit at a wavelength of 355/460 nm, as described in the Materials and Methods. Data represent means ± standard deviation (SD) (*n* = 6); * *p* < 0.05 vs. Con, # *p* < 0.05 vs. 24 h (GEF 100 mg/mL). GEF, gintonin-enriched fraction; GLP-1, glucagon-like protein-1.

**Figure 5 molecules-26-06298-f005:**
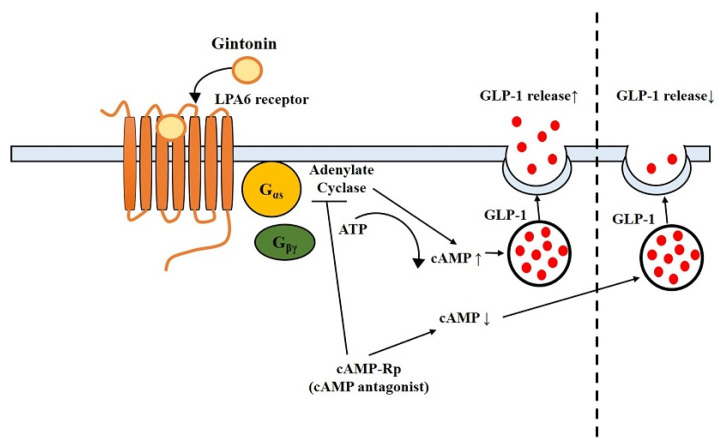
Schematic diagram of GEF-mediated GLP-1 release via LPA6-cAMP pathway. GEF, gintonin-enriched fraction; GLP-1, glucagon-like protein-1; LPA6, lysophosphatidic acid receptor 6; cAMP, cyclic adenosine monophosphate.

## Data Availability

Samples of the compounds are available from the authors.

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
