# Peer review of "Effect of the Gintonin-Enriched Fraction on Glucagon-Like-Protein-1 Release"

_molecules, 2021, doi:10.3390/molecules26206298_

Round 1

Reviewer 1 Report

The authors examined the expression of LPA receptors in the human colon cells expressions and found LPA6 is highly expressed, and cells respond to GEF via LPA6 receptors for GLP-1 secretion. The results appear to be interesting, but some points should be clarified.

  1. In Figure 1a and 1b, the author should have additional control cells, such as mouse L cells or other colon epithelial cells, to compare the expression profile of LPA receptors in NCI-H716 cells.
  2. In Figure 1b, the expression of LPA4 seems to be highly expressed in the blot as LPA6, although the quantitative results it is subtle. The authors should confirm this data to show more consistent results. 
  3. The author should provide the result of Ki16425 as a negative control in in vitro experiments but not only say it data not shown. 
  4. In Figure 3, knockdown of LPA6 expression by siRNA should be quantified. Also, what is the difference between Fig. 3a and 3b?
  5. In Figure S1, the authors should add LPA as a control.
  6. What is the amount or ratio of LPA in GEF?
  7. The level of GLP-1 secretion in mice was induced from 2.5 pM to ~3.0 pM after 50 mg/kg oral administration for 8h and 24h. Although there was a statistical significance, the differences were subtle. It will be more convincing to show more downstream indexes such as insulin secretion glucagon secretion to support this result. 
  8. Since LPA4 is highly expressed in NCI-H716 cells and both LPA4 and LPA6 can trigger cAMP pathway, it could not be exclusive the involvement of LPA4 in GLP-1 release. Therefore, the authors should use LPA4 antagonist and siRNA to validate the involvement of LPA. 
  9. It will be interesting to see the administration of agonists of LPA4 (AM966) and LPA6 (Alkyl OMPT) in vitro and in vivo has the same result as in GEF treatment. 

Reviewer 2 Report

Reviewers comments:-

The authors reported an absolute and interesting topic' titled' Effect of the Gintonin-Enriched fraction on Glucagon-Like Protein-1 Release. 

  • The work concluded by authors is captivating and the tone commands the reader to read and flow. I commend the authors for the valuable contribution. 
  • The scientific merit is sound  and clearly a continuous research stemming from other outputs. 
  • The overuse of the term 'We' in this manuscript is way too much, over 20 times the word 'we' was used. I strongly recommended that the authors significantly reduce and remove and start the sentences properly without 'we'. Lines, 19, 20, 73, 87, 101, 110, 115, 125, 142, 143,149,165, 166, 179, 183, 185, 186, 199, 208, 215, 236, 255 start with 'we'
  • Line 37, after the sentence ..gintonin rather enter references in which is/are included [1-5]
  • The authors tend to use difference tenses, I noticed past tense, past participle tense etc. Line 48, kindly remove 'has'
  • line 52 remove 'have'
  • Line 81 remove 'have been' and replace by 'were'
  • Line 104, the authors use ...by Kim et al., which Kim as I checked the reference there is more than one Kim et al., as first author, either use reference consistently or tag the year.
  • Line 124, remove have shown, replace by 'showed'
  • Line 194, remove 'has'

Supplementary data: Again remove 'we' and simply start the sentence ' The cytotoxicity of GED was determined by ....

Round 2

Reviewer 1 Report

1. For the western blot in Figure 1, the authors should run the lysate samples of NCI-H716 cells and mouse intestine tissues on the same gel and present the blot if the antibodies are the same. Also, please provide the information of LPA receptor antibodies, including company and catalog number, in Materials and Methods.  

2. Can the authors see the same result, as in the reference paper (Effects of Gintonin-enriched fraction on the gene expression of six lysophosphatidic receptor subtypes. Journal of Ginseng Research 45 (2021) 583-590), that the expression of LPA6 can be elevated by GEF? If Yes, please provide evidence and indicate the time and dose.

3. In Figure 3, the GLP-1 secretion shows a clear dose-response to GEF. However, the same dose-response did not happen in the mouse model in Figure 4. How do the authors explain the high dose of GEF (100 mg/kg) administration suppresses the GLP-1 secretion in mice? Please discuss in the Discussion. 
